# Diagnostic Value of EUS-FNA in the Differential Diagnosis of Esophageal Strictures Lacking Typical Malignant Features

**DOI:** 10.3390/diagnostics15192470

**Published:** 2025-09-26

**Authors:** Keyi Zhang, Qi He, Yu Jin, Caihan Duan, Jun Liu, Chaoqun Han, Rong Lin

**Affiliations:** Division of Gastroenterology, Union Hospital, Tongji Medical College, Huazhong University of Science and Technology, Wuhan 430022, China

**Keywords:** EUS-FNA, esophageal stricture, esophageal carcinoma, differential diagnosis, endosonography

## Abstract

**Background:** Esophageal strictures lacking typical malignant endoscopic features present a significant diagnostic challenge, often mimicking malignancy on imaging while concealing their true nature under regular white-light endoscopy. This study evaluated the utility of EUS-FNA in the differential diagnosis of such indeterminate strictures. **Methods:** We retrospectively analyzed 38 patients with suspicious malignant esophageal strictures indicated by CT but lacking definite malignant features on initial white-light gastroscopy. All patients underwent EUS-FNA for definitive pathological diagnosis. Clinicopathological data, imaging reports, endoscopic mucosal features, and procedural outcomes were assessed. **Results:** Among all 38 patients suspected of esophageal cancer by CT scan, 30 of them had malignant cytology results, including ESCC, EAC, metastatic cancer, and esophageal lymphoma. A total of 8 patients had benign findings, including esophageal tuberculosis, fungal esophagitis, eosinophilic esophagitis, and esophageal varices. Critically, EUS-FNA identified benign entities, such as eosinophilic esophagitis and esophageal tuberculosis masquerading as malignancy. CT features and mucosal features are also summarized and analyzed. **Conclusions:** EUS-FNA is a powerful tool for diagnosing esophageal strictures lacking typical malignant features. It reliably differentiates malignancy from challenging benign mimics, preventing misdiagnosis and guiding appropriate therapy. Clinicians should maintain a high suspicion for both occult malignancy and rare benign conditions in such stenotic lesions.

## 1. Introduction

Esophageal carcinoma (EC) is the sixth leading cause of cancer-related mortality and the eighth most common type of cancer worldwide [1,2]. Due to the difficulty of early diagnosis and its aggressive natural history, the mortality rate of EC remains at high levels [3,4]. While often asymptomatic in early stages, dysphagia alone and/or unintentional weight loss are the most common presenting clinical manifestations [5].

Gastroscopy of EC patients typically reveals esophageal masses, strictures, and mucosal congestion with brittleness and bleeding, ulceration, or overt neoplasms [6]. Diagnosis is usually confirmed by direct mucosal biopsy. Current guidelines recommend that all esophageal strictures should be biopsied to exclude malignancy, with a minimum of 6–8 biopsies advised for suspicious lesions to optimize diagnostic yield, as sampling error remains a significant limitation of standard white-light gastroscopy [7,8,9]. However, in a subset of EC, mucosal changes can be inconspicuous or smooth, presented solely as an esophageal stricture under standard white-light gastroscopy, further increasing the risk of false-negative results with conventional forceps biopsy.

Endoscopic ultrasonography (EUS), extending beyond traditional white-light gastroscopy, enables detailed visualization of the esophageal wall layers, providing information on lesion location, size, echogenicity, and layer involvement, and is crucial for tumor staging. Crucially, EUS can be combined with fine needle aspiration (FNA) or fine-needle biopsy (FNB) to acquire tissue samples for pathological examination simultaneously [10].

Accurate differential diagnosis is further complicated by several benign esophageal conditions, such as peptic strictures, eosinophilic esophagitis, or rare benign tumors such as esophageal leiomyoma [11,12,13,14,15]. These entities mimicking malignancy necessitate precise diagnostic methods. EUS-guided tissue acquisition represents a powerful tool for evaluating suspicious esophageal wall thickening underlying such strictures [16,17].

However, studies concerning the application of EUS-FNA in the differential diagnosis of suspicious esophageal strictures are limited. This study aimed to evaluate the role of EUS-FNA in the diagnosis of suspicious malignant esophageal strictures indicated by CT scans without typical mucosal features of EC. We also present several notable cases of diverse esophageal diseases mimicking malignancy under standard gastroscopy that required EUS-guided tissue acquisition for definitive diagnosis, illustrating the technique’s utility in clinical practice.

## 2. Methods

### 2.1. The Subjects

We retrospectively analyzed the patients with esophageal strictures with imaging findings (CT scans, X-ray, MRI) suspicious for malignancy but lacking typical mucosal alterations for EC diagnosis at Union Hospital of Tongji Medical College of Huazhong University of Science and Technology between January 2024 and December 2024. A total of 38 patients who had a suspicious malignant esophageal stricture and received EUS-FNA were enrolled in this study (Figure 1). All patients presented with various degrees of dysphagia or weight loss. Rapid on-site evaluation (ROSE) was employed at the endoscopist’s discretion when available to obtain a histopathological diagnosis. The clinical manifestations, imaging examinations, gastroscopy results, and EUS-FNA results were recorded and analyzed. This study was conducted in accordance with the Declaration of Helsinki and was approved by the ethics committee of Wuhan Union Hospital. All patients were informed of the potential risk of EUS-FNA operation and provided a signed consent form. All patients agreed to be evaluated for the study and had access to the study data.

Inclusion criteria were as follows: (1) patients with esophageal stricture indicated by imaging examinations (CT, X-ray, etc.); (2) patients without a prior confirmed diagnosis of EC; (3) patients who received biopsy for the suspicious lesion prior to the EUS-FNA operation and yielded negative results; (4) patients who received further diagnostic evaluation at the study center; and (5) patients with complete diagnostic and treatment records available.

Exclusion criteria were as follows: (1) patients with coagulation disorders (PLT < 50,000/mm^3^, INR > 1.5); (2) patients with a prior confirmed diagnosis of EC; (3) patients with esophageal stricture accompanied by typical feature of EC, such as mucosal ulceration, brittle mucosa or EC suggested by previous biopsy; or (4) patients with contraindications to endoscopic operation (e.g., pregnant women, cardiac or pulmonary dysfunction, intolerable to ultrasound endoscopy).

Rejection criteria were as follows: (1) patients who were found not to meet the inclusion criteria or who met the exclusion criteria during the study period; (2) patients with poor adherence and inability to complete the follow-up visits; and (3) patients without complete postoperative records.

### 2.2. Endoscopic Gastroscopy and EUS-Guided Tissue Acquisition Process

Preoperative assessment included routine blood tests, coagulation function, electrocardiography, and thoracic CT to exclude contraindications, such as severe cardiopulmonary diseases, coagulopathy, or anesthesia-related risks. Before the endoscopic procedure, general anesthesia was administered to the patient using propofol by a professional anesthesiologist, and vital signs were continuously monitored using a patient monitor during the EUS-FNA procedure.

Initial white-light gastroscopy revealed no malignant features, such as abnormal color, texture, and bulges, yielding negative findings. After the endoscopist reviewed the patient’s application form, an endoscopic ultrasound (GF-UCT 260; Olympus Corporation, Tokyo, Japan) was inserted to comprehensively evaluate the thickened esophageal wall, including lesion dimensions, location, and echogenicity. Under continuous ultrasound guidance, a fine-needle aspiration (FNA) needle (Cook 19G, Cook Medical Inc., Bloomington, IN, USA) was used for tissue sampling. The needle was inserted into the lesion using the standard freehand technique, with the endoscope positioned to ensure optimal needle trajectory and avoid vascular structures. After removal of the stylet, consistent negative pressure was applied using a 5 mL syringe. The needle was then moved to-and-fro within the lesion for approximately 10–15 times to obtain an adequate sample.

Following sampling, the negative pressure was released before the needle was withdrawn to minimize contamination and specimen dispersal. The acquired sample was then expressed onto a glass slide.

All patients were monitored for 12 h for procedure-related complications after the EUS-FNA operation.

### 2.3. Sample Processing

When ROSE (Rapid On-Site Evaluation) was performed, direct smears were prepared for immediate assessment. The hemorrhagic suction components and needle core tissue were used to prepare alcohol-fixed slides and tissue blocks for additional studies. Cytological and histological analysis primarily employed hematoxylin and eosin (H&E) staining, with immunohistochemical (IHC) staining complemented if necessary.

### 2.4. Postoperative Follow-Up of the Patients

The final diagnosis was established based on pathological findings, surgical findings, or the gold standard of respective diseases (such as a positive fungal glucan assay for fungal esophagitis). For patients without a definite diagnosis, telephone follow-up was conducted to check the potential diagnosis and patients’ status.

### 2.5. Data Collection

Participant demographics, clinical manifestation, laboratory findings, imaging reports, gastroscopy reports (both white light and EUS), pathologic findings, and clinical course of the patients were collected.

### 2.6. Statistical Analysis

Statistical analyses were performed with SPSS software, version 26.0 (IBM SPSS, Armonk, NY, USA). *p* < 0.05 was considered to indicate a statistically significant difference. Continuous variable results were reported as means with/without standard deviation (SD). Dichotomous variables are shown as percentages with or without 95% confidence intervals (CIs). The χ^2^ test was used for comparisons of rates and corrected where necessary.

## 3. Results

### 3.1. General Characteristics of the Patients

A total of 38 patients (26 male and 12 female) were enrolled in the study, and the average age was 61.4 years. (median age = 62 years) (Figure 1). Contrast-enhanced thoracic CT scans performed on all 38 patients revealed suspicious esophageal wall thickening; however, typical features of EC, such as mucosal ulceration and bleeding, were absent. The final diagnoses, confirmed by fine-needle aspiration (FNA), surgical findings, or the gold standard (such as a positive fungal glucan test for fungal esophagitis) during subsequent follow-up are presented in Table 1.

Adequate tissue samples for pathological or cytological assessment were obtained for all patients. Cytological examination identified malignancy in 30 cases (78.95%) and benign findings (including esophageal tuberculosis, fungal esophagitis, eosinophilic esophagitis, esophageal varices, esophageal leiomyoma, and reflux esophagitis) in 8 cases (21.05%). Among the 30 cases of malignancy, esophageal squamous cell carcinoma (ESCC) was the most common cause, accounting for 25 cases (65.79%); moreover, there were 2 cases of esophageal adenocarcinoma (EAC), 2 cases of lung-derived metastatic cancer, and 1 case of esophageal lymphoma.

Notably, among 30 patients with confirmed malignancy, 20 cases (66.67%) had radiologically suspicious lymph nodes, and 19 cases (63.33%) had anemia (17 mild anemia and 2 moderate anemia). In contrast, among 8 cases of benign diseases, only 4 had radiologically suspicious lymph nodes (50%), and 1 case had anemia (12.5%), which may suggest that the presence of lymphadenopathy and anemia may serve as helpful clues when determining the nature of unknown esophageal lesions.

### 3.2. CT Characteristics of the Patients

All patients have received a contrast-enhanced CT scan before the EUS-FNA process, indicating a suspicious esophageal stricture. Among the 38 patients, 10 showed even enhancement of the lesion, 18 showed uneven enhancement, and 10 showed no obvious enhancement. The contrast-enhanced CT scan results are summarized in Table 2.

### 3.3. Different Characteristics of Mucosa in Benign and Malignant Diseases

Mucosal changes were carefully examined and recorded by the operator and another experienced endoscopist present during the gastroscopy examination. We have excluded patients with typical mucosal changes of EC, and only patients lacking typical mucosal features were included in this study and analyzed. We investigated the changes in the mucosa and their correlation with different types of esophageal diseases. In benign diseases, 87.5% of all cases demonstrated smooth esophageal mucosa, while in malignancies, 60.0% of all cases demonstrated rough mucosa (uneven, grainy, thickened, or mild edema). The distribution of different mucosa morphologies in different diseases is listed in Table 3. Smooth mucosa tended to indicate the benign nature of the lesions, although this result is not statistically significant. (*p* = 0.132).

### 3.4. Different Characteristics of the Mucosa in Different Types of Esophageal Cancer

We further investigated mucosal features in different types of esophageal cancer (ESCC and EAC). In ESCC, patients tended to have a coarser mucosa (68.00%), while EAC patients tended to have smooth mucosa or showed little change in mucosa texture (Table 4). However, due to the limited cases of EAC included in the study, the generalizability of the findings is constrained.

Besides the texture of the mucosa, color changes of the mucosa were also reported in several cases. Two cases reported yellowish mucosa, 1 case of lymphoma and 1 case of ESCC; four cases reported reddish mucosa, including 3 cases of ESCC and 1 case of esophageal tuberculosis; and 1 case of EAC presenting pallor mucosa.

### 3.5. Rare Esophageal Diseases Characterized by Severe Esophageal Stenosis

#### 3.5.1. Fungal Esophagitis

Most cases of fungal esophagitis were reported in patients with immune deficits, while cases in normal people were also reported. Candida albicans (*C. albicans*) is the most common pathogen [18]. Patients with fungal esophagitis present with odynophagia, dysphagia, and atypical chest pain [19]. Tissue biopsies and fungal cultures can distinguish fungal esophagitis from cancer. For most esophageal fungal esophagitis, patients may have a normal esophageal lumen but white bean curd-like substance attaching to the esophageal mucosa. Through cytology brushing and culture, the pathogen can be determined. However, in our case (Patient#10), the patient was presented with abnormal thickening and stricture at the entrance of the esophagus. CT enhancement scan discovered lymphadenopathy within the mediastinum, and no obvious enhancement of the lesion was reported. A positive fungal glucan assay result was reported later (D-glucan: 263.1 pg/mL, normal range: 0–100.5 pg/mL). After anti-fungal therapy, the D-glucan level was restored to a normal level, and the lumen diameter gradually turned wider to normal (Figure 2).

#### 3.5.2. Eosinophilic Esophagitis (EoE)

Dysphagia and food impaction are the most common manifestations of eosinophilic esophagitis (EoE) in adults, while up to 50% of adult patients initially presenting with food impaction have a final diagnosis of EoE [20]. The most common endoscopic features in adults with EoE include linear furrows (80%) and mucosal rings (64%) [21]. However, in our case (Patient#24), despite stricture, the esophagus still had a normal appearance with a smooth surface. CT enhancement revealed no significant lymphadenopathy or enhancement of the lesion. The patient had mild anemia. Endoscopic features of EoE esophagus can be subtle and neglected during endoscopy, so esophageal biopsies are necessary in all patients suspected of having EoE, irrespective of endoscopic appearance [22]. The patient had a significant increase in the eosinophil level in the routine blood test. (Eosinophil level: 0.76 G/L, normal: 0.02–0.52 G/L). Tissue acquired by EUS-FNA demonstrated cellulose-like exudate and several small patches of smooth muscle tissue with scattered eosinophilic infiltration within the smooth muscle (>15/HPF) (Figure 3).

#### 3.5.3. Esophageal Tuberculosis

Esophageal tuberculosis (ET) is very rare. Most of the reported cases have been secondary to pulmonary tuberculosis [13]. Unexplained dysphagia and upper gastrointestinal bleeding are the most common symptoms of ET. Esophageal tuberculosis may present in three forms at upper gastrointestinal endoscopy: ulcerative, hyperplastic, or granular. The hypertrophic form occurs as a consequence of fibrosis of the esophageal wall with a pseudotumoral presentation, which can be difficult to distinguish from a malignancy [23]. In our case (Patient#29), the patient was presented with intermittent dysphagia and progressive exacerbation. The patient had a previous history of pulmonary tuberculosis. CT enhancement scan indicated a posterior mediastinal mass, which had no clear border with the esophagus and was combined with necrosis. No significant lymphadenopathy or enhancement of the lesion was discovered. Samples acquired from EUS-FNA showed granulomatous inflammation with necrosis, consistent with the presentation of tuberculosis. The mycobacterium tuberculosis nucleic acid test result of the acquired sample was positive, and the T-spot test of the patient was also positive, consistent with the final diagnosis of esophageal tuberculosis (Figure 4).

#### 3.5.4. Primary Esophageal MALT Lymphoma

MALT (mucosa-associated lymphoid tissue) lymphoma of the esophagus is a rare entity that accounts for less than 1% of all esophageal tumors [24]. MALT lymphoma is a low-grade malignant B-cell lymphoma that is mainly seen at the mucosal sites, which possesses marginal zone B cells. Its nonspecific clinical presentation and variable histological features made it difficult to make a definite diagnosis. The most frequent complaint of esophageal MALT lymphoma is dysphagia due to esophageal stricture. In our case (Patient#31), the patient was presented with progressive dysphagia for 1 year and significant weight loss (decreased by 15 kg in one year). CT enhancement scan indicated a thickened esophagus wall from the entrance to the lower part of the esophagus, with moderate homogeneous enhancement, and narrowing of the lumen, which is evident in the middle and upper part of the esophagus, and we suspected a tumor lesion. No significant lymphadenopathy was discovered. EUS indicated a submucosal, inhomogeneous, hypoechoic lesion with still clear borders and little blood flow signal. The final diagnosis was made by EUS-FNA: primary mucosa-associated lymphoid tissue lymphoma (Figure 5).

## 4. Discussion

Differential diagnosis of esophageal malignancy and other esophageal diseases sharing similar features (such as esophageal tuberculosis, eosinophilic esophagitis, etc.) necessitates precise diagnostic methods. Compared to CT scans or other medical imaging methods, the advent of EUS has significantly enhanced the diagnostic accuracy of esophageal diseases, including staging of esophageal cancer, evaluation of esophageal stricture, etc. EUS can provide detailed information about the lesion, including the level of tumor origin, echogenicity, borders, blood flow signal, etc. Moreover, EUS-guided sample acquisition, including EUS-FNA, EUS-FNB, or biopsy by forceps, has combined histopathological or cytological examination for precise diagnosis of esophageal diseases presenting esophageal stricture. A recent network meta-analysis focusing on solid pancreatic masses demonstrated that specific techniques like modified wet suction provide superior histological yield compared to conventional methods, being associated with higher sample adequacy and tissue integrity, which may provide clues for further enhancing the diagnostic accuracy of EUS-guided sampling [25]. Although most evidence originates from pancreatic studies, these findings regarding the technical approach may also optimize diagnostic performance during EUS-guided sampling of esophageal lesions. This principle is further supported by the very recent European Society of Gastrointestinal Endoscopy (ESGE) Technical and Technology Review [26], which provides comprehensive evidence-based recommendations on EUS-guided tissue sampling, underscoring the critical impact of needle selection and technique on diagnostic outcomes.

While EUS-FNA was central to our diagnostic approach, it is not universally indicated. In our practice, EUS is primarily reserved for strictures with concerning features on cross-sectional imaging (e.g., wall thickening, extrinsic compression) or those without a conclusive diagnosis after initial conventional endoscopy. Situations where we might defer EUS include obvious intraluminal masses with easily assessable mucosal abnormalities, in which extensive forceps biopsies remain the first-line tissue acquisition method. Furthermore, for strictly superficial or mucosal-based lesions where submucosal involvement is not suspected, advanced endoscopic resection techniques, such as endoscopic mucosal resection (EMR), could be considered both diagnostic and therapeutic. This decision-making process emphasizes that the choice of technique should be individualized based on endoscopic appearance, clinical suspicion, and available expertise.

Despite these advantages, certain benign and malignant esophageal conditions remain challenging to distinguish under conventional endoscopy, particularly when typical features such as ulceration or bleeding are absent. In such cases, EUS-guided sampling proves invaluable. In our study, all 38 patients achieved a definitive diagnosis through these techniques.

Critically, among all 38 patients who had esophageal stricture but lacked diagnostic evidence for esophageal cancer, 30 were confirmed with esophageal malignancies. This underscores the imperative for clinicians to maintain a high index of suspicion for malignancy in such scenarios. In addition, for some of the cases included in this study, these conditions are very rare (such as esophageal tuberculosis and esophageal MALT lymphoma), which also necessitates that clinicians maintain awareness of these entities to facilitate accurate diagnosis and avoid misdiagnosis as EC.

This study has several limitations that should be acknowledged. First, its retrospective and single-center design introduces potential selection bias and may limit the generalizability of our findings. Second, the sample size was relatively small (only 38 cases), particularly for rare benign conditions and the EAC subgroup, which reduced the statistical power for meaningful subgroup comparisons. Future prospective, multi-center studies with larger cohorts are needed to validate our results. Third, the use of ROSE and the selection of needles were not standardized, which may have affected diagnostic performance despite all cases yielding adequate samples.

## 5. Conclusions

In conclusion, EUS-FNA demonstrates exceptional diagnostic accuracy in evaluating esophageal strictures lacking typical malignant endoscopic features. It is indispensable not only for confirming malignancy but also for identifying rare benign mimics, thereby guiding appropriate management and preventing misdiagnosis.

## Figures and Tables

**Figure 1 diagnostics-15-02470-f001:**
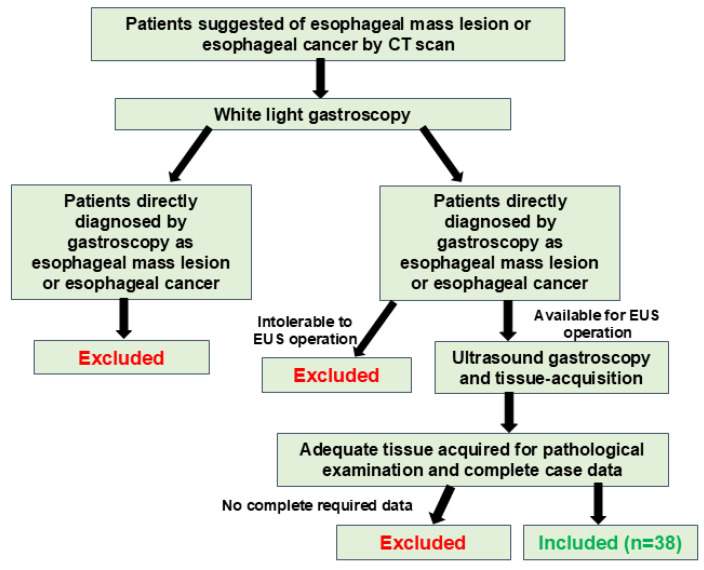
Schematic chart of the patient enrollment process.

**Figure 2 diagnostics-15-02470-f002:**
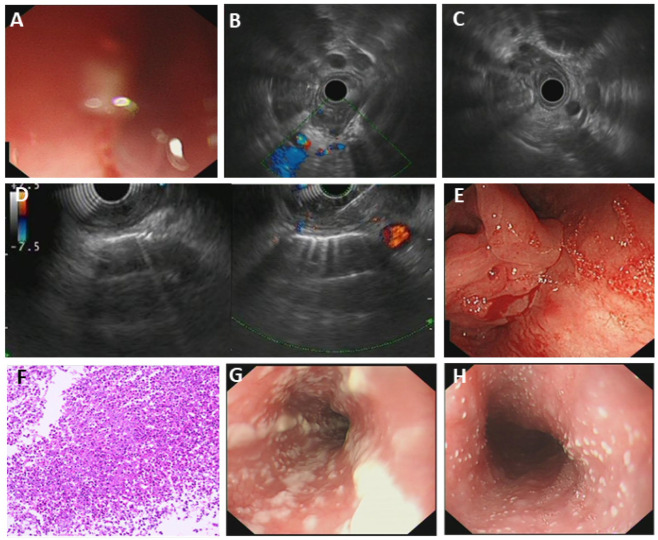
Fungal esophagitis. (**A**) White-light gastroscopy indicating stricture at the entrance of the esophagus; (**B**,**C**) endoscopic ultrasonography (EUS) was performed at the stricture of the esophageal entrance, and the whole layer of esophageal wall was found to be hypoechoic and thickened; (**D**) EUS-FNA was performed; (**E**) the replacement of the gastroscope found that the puncture site was located near the entrance of the esophagus, and a little blood seepage; (**F**) histopathology indicated fungal infection; (**G**,**H**) common fungal infection of the esophagus showing no conspicuous changes of the esophageal lumen, but we can see white bean curd-like substance attaching to the mucosa.

**Figure 3 diagnostics-15-02470-f003:**
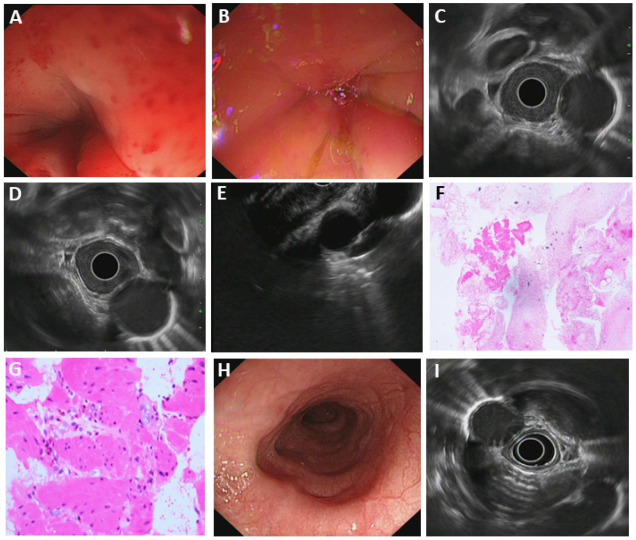
Eosinophilic esophagitis. (**A**,**B**) White-light gastroscopy; (**C**–**E**) EUS gastroscopy; (**F**,**G**) histopathology: tissue acquired by EUS-FNA demonstrated cellulose-like exudate and several small patches of smooth muscle tissue with scattered eosinophilic infiltration within the smooth muscle (>15/HPF); (**H**) white-light gastroscopy after treatment, the stricture was significantly relieved; (**I**) EUS gastroscopy after treatment.

**Figure 4 diagnostics-15-02470-f004:**
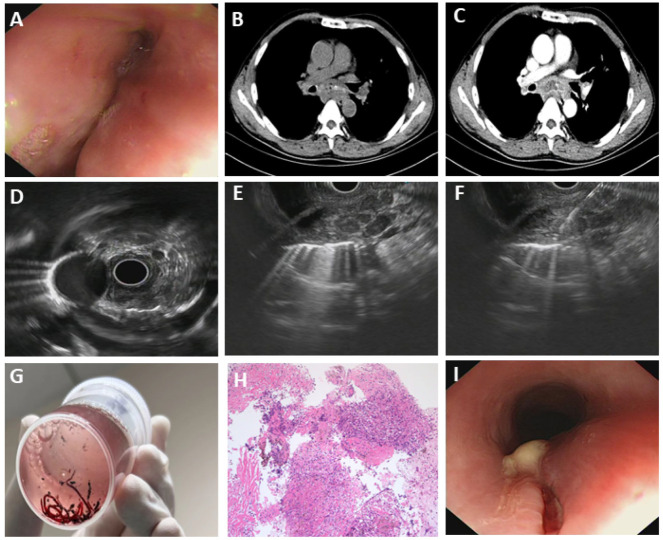
Esophageal tuberculosis. (**A**) White-light gastroscopy; (**B**,**C**) CT scans and CT enhancement scans indicated posterior mediastinal mass, which had no clear border with the esophagus and was combined with necrosis; (**D**–**F**) EUS gastroscopy; (**G**) tissue sample acquired by EUS-FNA; (**H**) EUS-FNA acquired tissue indicated granulomatous inflammation with necrosis and positive for mycobacterium tuberculosis nucleic acid test, supporting the diagnosis of esophageal tuberculosis. (**I**) White substance attached to esophageal mucosa.

**Figure 5 diagnostics-15-02470-f005:**
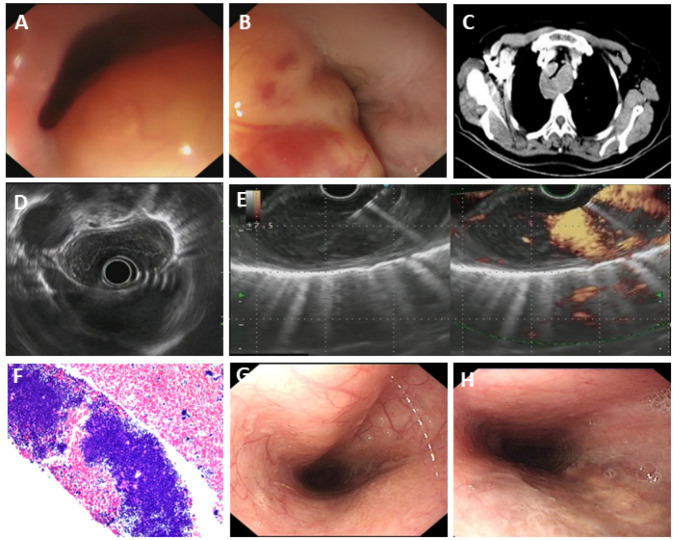
Primary esophageal MALT lymphoma. (**A**,**B**) White-light gastroscopy of different parts of the thickened esophageal wall; (**C**) CT enhancement scan indicated thickening of the esophageal wall; (**D**,**E**) EUS gastroscopy of the lesion; (**F**) histopathology: primary mucosa-associated lymphoid tissue lymphoma (MALT); (**G**,**H**) after chemotherapy, the stricture extent greatly decreased, and the manifestation of dysphagia of the patient was significantly relieved.

**Table 1 diagnostics-15-02470-t001:** General characteristics, lesion location, and the final diagnoses of the patients.

No.	Sex *	Age	Lesion Location	Final Diagnosis	Presence and Location of Lymphadenopathy	Presence and Degree of Anemia **
**1**	**F**	45	Mid-thoracic	Lung-derived metastatic cancer	Bilateral cervical lymph nodes	-
**2**	**M**	62	Mid-thoracic	Lung-derived metastatic cancer	-	-
**3**	**F**	72	Lower thoracic	Esophageal adenocarcinoma	Bilateral axillary	Mild
**4**	**M**	65	Upper-thoracic	Esophageal squamous cell carcinoma	Mediastinum and right hilar	-
**5**	**M**	75	Upper-thoracic	Esophageal squamous cell carcinoma	Bilateral hilar	-
**6**	**M**	47	Mid-thoracic	Esophageal squamous cell carcinoma	-	Moderate
**7**	**M**	63	Mid-thoracic	Esophageal squamous cell carcinoma	-	-
**8**	**M**	73	Upper-thoracic	Esophageal squamous cell carcinoma	Mediastinum	Mild
**9**	**F**	52	Mid-thoracic	Esophageal squamous cell carcinoma	Right hilar	-
**10**	**F**	52	Mid-thoracic	Fungal esophagitis	Mediastinum	-
**11**	**M**	52	Mid-thoracic	Esophageal squamous cell carcinoma	-	Mild
**12**	**M**	71	Lower-thoracic	Esophageal squamous cell carcinoma	Right hilar	Mild
**13**	**M**	82	Cervical	Esophageal squamous cell carcinoma	-	Mild
**14**	**M**	63	Upper-thoracic	Esophageal squamous cell carcinoma	Bilateral supraclavicular	Mild
**15**	**F**	43	Upper-thoracic	Gastroesophagitis	-	-
**16**	**M**	70	Mid-thoracic	Esophageal squamous cell carcinoma	Mediastinum	-
**17**	**M**	75	Upper-thoracic	Esophageal squamous cell carcinoma	Mediastinum	Mild
**18**	**M**	70	Lower-thoracic	Esophageal adenocarcinoma	Peri-esophageal and mediastinum	Mild
**19**	**M**	61	Upper-thoracic	Esophageal squamous cell carcinoma	Hepatogastric ligament	Mild
**20**	**F**	37	Cervical	Esophageal varices	-	-
**21**	**M**	55	Upper-thoracic	Esophageal squamous cell carcinoma	Mediastinum	-
**22**	**M**	52	Mid-thoracic	Esophageal squamous cell carcinoma	-	Mild
**23**	**M**	77	Lower-thoracic	Esophageal squamous cell carcinoma	Mediastinum and bilateral hilar	Mild
**24**	**F**	55	Mid-thoracic	Eosinophilic esophagitis	-	Mild
**25**	**M**	62	Mid-thoracic	Esophageal squamous cell carcinoma	Mediastinum and bilateral hilar	Mild
**26**	**F**	57	Lower-thoracic	Esophageal squamous cell carcinoma	Mediastinum and bilateral hilar	-
**27**	**F**	62	Lower-thoracic	Esophageal leiomyoma	Hepatogastric ligament	-
**28**	**F**	39	Lower-thoracic	Esophageal leiomyoma	Mediastinum and bilateral axillary	-
**29**	**M**	47	Mid-thoracic	Esophageal tuberculosis	-	-
**30**	**F**	63	Mid-thoracic	Esophageal squamous cell carcinoma	-	Mild
**31**	**M**	62	Full-length	Esophageal lymphoma	-	-
**32**	**M**	65	Upper-thoracic	Esophageal squamous cell carcinoma	-	Mild
**33**	**F**	69	Lower-thoracic	Esophageal squamous cell carcinoma	Left hilar and mediastinum	Mild
**34**	**M**	68	Full-length	Achalasia	Mediastinum	-
**35**	**M**	65	Mid-thoracic	Esophageal squamous cell carcinoma	-	Mild
**36**	**M**	77	Mid-thoracic	Esophageal squamous cell carcinoma	Mediastinum	Mild
**37**	**M**	63	Cervical	Esophageal squamous cell carcinoma	Mediastinum	-
**38**	**M**	67	Lower-thoracic	Esophageal squamous cell carcinoma	Mediastinum	Moderate

* M = Male, F = Female. ** Anemia was defined and graded according to WHO criteria as follows: for adults excluding pregnant adult females, Hb ≤ 120 g/L; mild anemia: Hb 90–119 g/L; moderate anemia: Hb 60–89 g/L; severe anemia: Hb 30–59 g/L; and very severe anemia, Hb < 30 g/L.

**Table 2 diagnostics-15-02470-t002:** Contrast-enhanced CT features of the patients.

Final Diagnosis	Contrast-Enhanced CT Enhancement Feature *		
	Uneven Enhancement	Even Enhancement	No Significant Enhancement
**Esophageal** squamous cell carcinoma **(ESCC)**	13/25 (52%)	7/25 (28%)	5/25 (20%)
**Esophageal adenocarcinoma (EAC)**	0/2 (0%)	1/2 (50%)	1/2 (50%)
**Metastatic cancer**	2/2 (100%)	0/2 (0%)	0/2 (0%)
**Fungal esophagitis**	1/1 (100%)	0/1 (0%)	0/1 (0%)
**Gastroesophagitis**	0/1 (0%)	1/1 (100%)	0/1 (0%)
**Eosinophilic esophagitis**	0/1 (0%)	0/1 (0%)	1/1 (100%)
**Achalasia**	0/1 (0%)	0/1 (0%)	1/1 (100%)
**Esophageal lymphoma**	0/1 (0%)	1/1 (100%)	0/1 (0%)
**Esophageal tuberculosis**	0/1 (0%)	0/1 (0%)	1/1 (100%)
**Esophageal varices**	1/1 (100%)	0/1 (0%)	0/1 (0%)
**Esophageal leiomyoma**	1/2 (50%)	0/2 (0%)	1/2 (50%)

* Example of uneven enhancement.

**Table 3 diagnostics-15-02470-t003:** Mucosa texture change in benign and malignant diseases.

Texture of Mucosa *	Coarse	Smooth
**Benign diseases**	1/8 (12.5%)	7/8 (87.5%)
**Malignant diseases**	15/30 (50%)	15/30 (50%)

* Examples of a “coarse” and “smooth” mucosa, respectively (see Appendix A).

**Table 4 diagnostics-15-02470-t004:** Mucosal features in different types of esophageal cancer.

	Coarse Mucosa	Smooth Mucosa
**Esophageal squamous cell carcinoma (ESCC)**	17/25 (68%)	8/25 (32%)
**Esophageal Adenocarcinoma (EAC)**	0/2 (0%)	2/2 (100%)

## Data Availability

The data presented in this study are available on request from the corresponding author due to the sensitive nature of the human subject data.

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
