# Peer review of "Diagnostic Value of EUS-FNA in the Differential Diagnosis of Esophageal Strictures Lacking Typical Malignant Features"

_diagnostics, 2025, doi:10.3390/diagnostics15192470_

Round 1

Reviewer 1 Report

Comments and Suggestions for Authors

The topic is important and the technique relatively new. My comments:

1) THe authors shoudl explain more procedural details. Which needle size? FNA or FNB? How was the needle inserted? Which sampling technique? 

2) Following the point above, the authors should improve the discussion (which is too short!). They should mention the current evidence on the needles and techniques for tissue sampling. Although most of these evidence are based on RCTs in pancreatic masses, they can be applicable to EUS-sampling in general (cite the relevant NMA: PMID: 36657607)

3) Again, the discussion can be improved considering the current literature, for example the very recent ESGE guidelines on EUS-tissue sampling

4) The retrospective design and the limited sample size represent major limitations and they should be addressed as such in the Discussion

5) Please spell out all the abbreviations in the Tables

Author Response

#Reviewer 1

Q1. The authors should explain more procedural details. Which needle size? FNA or FNB? How was the needle inserted? Which sampling technique?

A1: We would thank the reviewer for this kind comment. We agree that a more detailed description of the technical procedure is essential for reproducibility and clarity. We have now thoroughly revised the Methods section (specifically, the procedural subsection) to incorporate these details. The revisions can be found in Page3, Line 106-125, detailed as:

“Before the endoscopic procedure, general anesthesia was administered to the patient using propofol by a professional anesthesiologist, and the vital signs were continuously monitored using a patient monitor during the procedure.

Initial white-light gastroscopy revealed no malignant features, such as abnormal color, texture, and bulges, yielding negative findings. After the endoscopist reviewed the patient’s application form, an endoscopic ultrasound (GF-UCT 260; Olympus Cor-poration, Tokyo, Japan) was inserted to comprehensively evaluate the thickened esophageal wall, including lesion dimensions, location, and echogenicity. Under con-tinuous ultrasound guidance, a fine-needle aspiration (FNA) needle (Cook 19G, Cook Medical Inc., Bloomington, IN, United States) was used for tissue sampling. The needle was inserted into the lesion using the standard freehand technique, with the endoscope positioned to ensure optimal needle trajectory and avoid vascular structures. After re-moval of the stylet, consistent negative pressure was applied using a 5-mL syringe. The needle was then moved to-and-fro within the lesion for approximately 10-15 times to obtain an adequate sample.

Following sampling, the negative pressure was released before the needle was withdrawn to minimize contamination and specimen dispersal. The acquired sample was then expressed onto a glass slide.

All patients were monitored for 12 hours for procedure-related complications after the EUS-FNA operation.”

  1. Following the point above, the authors should improve the discussion (which is too short!). They should mention the current evidence on the needles and techniques for tissue sampling. Although most of this evidence are based on RCTs in pancreatic masses, they can be applicable to EUS-sampling in general. (cite the relevant NMA: PMID: 36657607)

A2: We thank the reviewer for this constructive comment on our paper. According to your suggestion, we have checked current literature and cited relative literature wherever appropriate. We also have cited the paper that you have suggested (reference number: 25), as it helped us to further refine and strengthen the discussion in our paper. We have also revised the Discussion section of our manuscript, the revisions can be found in Page 7, Line 279-285, detailed as:

“ A recent network meta-analysis focusing on solid pancreatic masses demonstrated that specific techniques like modified wet suction provide superior histological yield compared to conventional methods, being associated with higher sample adequacy and tissue integrity, which may provide clues for further enhancing the diagnostic accuracy of EUS-guided sampling.(25) Although most evidence originates from pancreatic stud-ies, these findings regarding technical approach may also optimize diagnostic perfor-mance during EUS-guided sampling of esophageal lesions. ”

Q3. Again, the discussion can be improved considering the current literature, for example the very recent ESGE guidelines on EUS-tissue sampling.

A3: Thank you for this constructive advice. We have checked recent literature and the relevant ESGE guideline on EUS-tissue sampling and revised the Discussion part of our manuscript. We believe the additional citation of the ESGE guidelines can help to strengthen the persuasiveness and raise the quality of our study. (reference number: 26) The revisions can be found in Page7, Line 286-289, detailed as:

“ This principle is further supported by the very recent European Society of Gastroin-testinal Endoscopy (ESGE) Technical and Technology Review (26), which provides comprehensive evidence-based recommendations on EUS-guided tissue sampling, un-derscoring the critical impact of needle selection and technique on diagnostic out-comes.”

Q4. The retrospective design and the limited sample size represent major limitations and they should be addressed as such in the Discussion.

A4: We would thank the reviewer for this important comment. According to your suggestion, we have revised our Discussion and illustrated the limitations of our study. The revisions can be found in Page8, Line 313-320, detailed as:

“This study has several limitations that should be acknowledged. First, its retro-spective and single-center design introduces potential selection bias and may limit the generalizability of our findings. Second, the sample size was relatively small (only 38 cases), particularly for rare benign conditions and the EAC subgroup, which reduced the statistical power for meaningful subgroup comparisons. Future prospective, mul-ti-center studies with larger cohorts are needed to validate our results. Third, the use of ROSE and the selection of needles were not standardized, which may have affected di-agnostic performance despite all cases yielding adequate samples. ”

Q5. Please spell out all the abbreviations in the Tables.

A5: Thank you for this important comment. We have complemented the full spellings of the abbreviations in the tables as suggested.

Reviewer 2 Report

Comments and Suggestions for Authors

Thanks for your submission which I found interesting to read.

I would like to commend you on your use of EUS in differentiating the cause of oesophageal stricture - this is an important area and something that requires a high level of expertise.

My main critique is around how the reader can learn maximally from your experiences and I feel there is more to be discussed here:

  1. In your introduction you mention the difficulty in standard gastroscopy in diagnosing oesophageal strictures - might I suggest you include some references on: a) the need that all strictures are biopsied, and b) if suspiscious of malignancy, all strictures probably need at least 6 biopsies (some data suggests 8 or even more). This is an important point as sometimes the diagnostic yield of standard gastroscopy is limited by the endoscopist simply not taking enough biopsies
  2. In your methods, you mention how you perform EUS. Can you please expand regarding what sedation you use (I assume propofol or other form of general anaesthesia) - this is helpful as not all countries in the world use propofol and I can see you have some very high strictures in your case series which simply would not be possible to EUS FNA with standard doses of midazolam/fentanyl.
  3. Did any patients have standard biopsies taken before EUS and if so, what was the yield compared to EUS?
  4. I would like some extra detail for the cases - in particularly in differentiating between benign and malignant, do you have any detail on whether there were any local lymph nodes seen on CT scan, or whether the patients had any evidence of anaemia...we have found in our own experience that these are helpful clues
  5. Table 2. I would like more useful information for the reader here. For example, are you able to provide some images to demonstrate what even enhancement and uneven enhancement looks like as a panel alongside this table
  6. Similarly - Table 3 - I think it would be very helpful for you to show endoscopic images to show what you mean by rough and smooth. We know that malignant disease usually looks rough, but perhaps the main role of this paper here should be in helping the reader to understand what a smooth malignancy looks like. 
  7. Table 4 - same as with Table 3: having example images of what you define as smooth and rough mucosa would be helpful.
  8. All Tables: Please include both number of cases and percentage, e.g. 15/30 (50%)
  9. I don't think the figure legends of figures 3 and match the images, eg: Figure 3F says white light endoscopy but the image is a pathology slide, H-I says histopathology but H shows an endoscopic image and I shows an EUS image. Fig 4 B-C are CT images yet are labelled as EUS gastroscopy, D-E are EUS images but are labelled as CT scans. Please carefully check all figures and amend as appropriate
  10. Do you have some data on where you used FNA or FNB, decision making behind this?
  11. In your discussion, it would be helpful for the authors to discuss where they may use EUS (I presume most cases) but also if there are any situations where they wouldn't use EUS and what decision making would involve in such circumstances - eg would they consider more biopsies, bite on bite biopsies, an endoscopic mucosal resection for more tissue for analysis, etc. etc. 

Many thanks.

Author Response

#Reviewer 2

Q1: In your introduction you mention the difficulty in standard gastroscopy in diagnosing esophageal strictures - might I suggest you include some references on: a) the need that all strictures are biopsied, and b) if suspicious of malignancy, all strictures probably need at least 6 biopsies (some data suggests 8 or even more). This is an important point as sometimes the diagnostic yield of standard gastroscopy is limited by the endoscopist simply not taking enough biopsies.

A1: Thank you for this insightful comment. Among the cases we included, most of the surface mucosa was smooth or coarse and the lesions were located beneath the mucosal layer. Therefore, the biopsies were all negative and endoscopic ultrasound-guided biopsy was required.

We greatly appreciate your insightful comment regarding the critical importance of obtaining an adequate number of biopsies during standard gastroscopy to optimize the diagnostic yield for esophageal strictures. We fully agree with your point, and we have revised the introduction accordingly to incorporate authoritative references that support both the necessity of biopsying all strictures and the requirement for multiple biopsies when malignancy is suspected. According to your suggestion, we have referred to relevant literature, and we have revised our Introduction section, the revisions can be found in Page2, Line46-52, detailed as:

“Current guidelines recommend that all esophageal strictures should be biopsied to ex-clude malignancy, with a minimum of 6-8 biopsies advised for suspicious lesions to op-timize diagnostic yield, as sampling error remains a significant limitation of standard white-light gastroscopy. (7-9) However, in a subset of EC, mucosal changes can be in-conspicuous or smooth, presented solely as an esophageal stricture under standard white-light gastroscopy, further increasing the risk of false-negative results with con-ventional forceps biopsy.”

Q2: In your methods, you mention how you perform EUS. Can you please expand regarding what sedation you use (I assume propofol or other form of general anesthesia) - this is helpful as not all countries in the world use propofol and I can see you have some very high strictures in your case series which simply would not be possible to EUS FNA with standard doses of midazolam/fentanyl.

A2: We thank the reviewer for the opportunity to further clarify this important point. The sedation we applicated was propofol and the whole sedation process was supervised by professional anesthesiologist. We would complement this detail in our revised manuscript, the revisions can be found in Page3, Line107-109, detailed as:

“Before the endoscopic procedure, general anesthesia was administered to the patient using propofol by a professional anesthesiologist, and the vital signs were continuously monitored using a patient monitor during the EUS-FNA procedure. ”

Q3: Did any patients have standard biopsies taken before EUS and if so, what was the yield compared to EUS?

A3: Thank you for your question. We retrospectively analyzed 38 patients with suspicious malignant esophageal strictures indicated by CT but lacking definite malignant features on initial white-light gastroscopy. Among the cases we included, most of the surface mucosa was smooth or rough and the lesions were located beneath the mucosal layer. Therefore, the biopsies were all negative and endoscopic ultrasound-guided biopsy was required. All patients underwent EUS-FNA for definitive pathological diagnosis. To further clarify these details ,we have complemented a statement regarding this in our Methods section (Page2-3, Line89-90), detailed as:

“Inclusion criteria were as follows: (1) Patients with esophageal stricture indicated by imaging examinations (CT, X‑ray, etc.); (2) Patients without a prior confirmed diag-nosis of EC; (3) Patients who received biopsy for the suspicious lesion prior to the EUS-FNA operation and yielded negative results; (4) Patients who received further di-agnostic evaluation at the study center; (5) Patients with complete diagnostic and treatment records available.”

Q4: I would like some extra detail for the cases - in particularly in differentiating between benign and malignant, do you have any detail on whether there were any local lymph nodes seen on CT scan, or whether the patients had any evidence of anemia...we have found in our own experience that these are helpful clues.

A4: We sincerely thank the reviewer for this constructive comment. We have checked the CT scan report and the blood routine result again, and we found that these clues rather significant. We have complemented this information in the manuscript and the revised Table1, and we also added discussion regarding these issues in our revised manuscript, detailed in each cases we presented in our manuscript.

Q5: Table 2. I would like more useful information for the reader here. For example, are you able to provide some images to demonstrate what even enhancement and uneven enhancement looks like as a panel alongside this table.

A5: Thank you for this advice. We have now included a representative EUS image demonstrating a clear case of uneven enhancement as a new Supplementary Figure 1. We have also added a citation to this figure in the caption of Table 2 to directly guide the reader to this visual example. Unfortunately, due to limited access to the imaging database, we were unable to identify a high-quality, clear example of even enhancement from a benign esophageal stricture that was suitable for publication. This is partly because truly 'even' enhancement can be subtle and less visually striking. We believe that the addition of this supplementary image, prompted by the reviewer's insightful comment, significantly strengthens the manuscript. We are grateful for this guidance. While we regret that we cannot provide a perfect paired comparison, we hope that the provided example of uneven enhancement will serve as a valuable reference point for readers.

Q6: Similarly - Table 3 - I think it would be very helpful for you to show endoscopic images to show what you mean by rough and smooth. We know that malignant disease usually looks rough, but perhaps the main role of this paper here should be in helping the reader to understand what a smooth malignancy looks like.

A6: Thank you for this advice. We have complemented images and respective figure legends (Supplementary figure S2) as you proposed, and the complemented figures were presented as a panel under Table 3.

Q7: Table 4 - same as with Table 3: having example images of what you define as smooth and rough mucosa would be helpful.

A7: Thank you again for this advice. We have complemented images and respective figure legends (Supplementary figure S2) as you proposed, and the complemented figures were presented as a panel under Table 3.

Q8: All Tables: Please include both number of cases and percentage, e.g. 15/30 (50%)

A8: Thank you for this important comment. We have included the information as you suggested in the tables.

Q9: I don't think the figure legends of figures 3 and match the images, e.g.: Figure 3F says white light endoscopy but the image is a pathology slide, H-I says histopathology but H shows an endoscopic image and I shows an EUS image. Fig 4 B-C are CT images yet are labelled as EUS gastroscopy, D-E are EUS images but are labelled as CT scans. Please carefully check all figures and amend as appropriate.

A9: We thank the reviewer for allowing us to clarify this point. We have revised the figure legends as you proposed. Thank you very much! The revised figure are marked with red fonts under the respective figures, specifically, Figure 3 and 4.

Q10: Do you have some data on where you used FNA or FNB, decision making behind this?

A10: Thank you very much for your kind consideration. We retrospectively analyzed 38 patients with suspicious malignant esophageal strictures indicated by CT but lacking definite malignant features on initial white-light gastroscopy. Among the cases we included, FNA needles (Cook 19G, Cook Medical Inc., Bloomington, IN, United States) were used for tissue sampling. The needle was inserted into the lesion using the standard freehand technique, with the endoscope positioned to ensure optimal needle trajectory and avoid vascular structures. FNB needles are not routinely stocked in our hospital, and the positive rate of the FNA needles is also very high in our hospital. Among the 38 patients with smooth surfaces or negative biopsy results, all were finally diagnosed through EUS-FNA. Additionally, according to recent literature, the diagnostic accuracy of EUS-FNA and EUS-FNA can be comparable. (PMID: 34071881)

Q11: In your discussion, it would be helpful for the authors to discuss where they may use EUS (I presume most cases) but also if there are any situations where they wouldn't use EUS and what decision making would involve in such circumstances - e.g. would they consider more biopsies, bite on bite biopsies, an endoscopic mucosal resection for more tissue for analysis, etc. etc.

A11: Thank you for this valuable comment. We have complemented relevant discussion in our revised Discussion section of the manuscript, the revisions can be found in Page7, Line 290-300, detailed as:

“While EUS-FNA was central to our diagnostic approach, it is not universally indi-cated. In our practice, EUS is primarily reserved for strictures with concerning features on cross-sectional imaging (e.g. wall thickening, extrinsic compression) or those with-out a conclusive diagnosis after initial conventional endoscopy. Situations where we might defer EUS exclude obvious intraluminal masses with easily assessible mucosal abnormalities, in which extensive forceps biopsies remain the first-line tissue acquisi-tion method. Furthermore, for strictly superficial or mucosal-based lesions where sub-mucosal involvement is not suspected, advanced endoscopic resection techniques, such as endoscopic mucosal resection (EMR), could be considered both diagnostic and ther-apeutic. This decision-making process emphasizes that the choice of technique should be individualized based on endoscopic appearance, clinical suspicion and available ex-pertise.”

Round 2

Reviewer 1 Report

Comments and Suggestions for Authors

The revised manuscript is OK